# Application of Biopsy Samples Used for *Helicobacter pylori* Urease Test to Predict Epstein–Barr Virus-Associated Cancer

**DOI:** 10.3390/microorganisms8060923

**Published:** 2020-06-18

**Authors:** Andy Visi Kartika, Hisashi Iizasa, Dan Ding, Yuichi Kanehiro, Yoshitsugu Tajima, Shunsuke Kaji, Hideo Yanai, Hironori Yoshiyama

**Affiliations:** 1Department of Microbiology, Faculty of Medicine, Shimane University, 89-1 Enya, Izumo, Shimane 693-8504, Japan; vkartika@med.shimane-u.ac.jp (A.V.K.); iizasah@med.shimane-u.ac.jp (H.I.); m199605@med.shimane-u.ac.jp (D.D.); kanehiro@med.shimane-u.ac.jp (Y.K.); s.kaji@med.shimane-u.ac.jp (S.K.); 2Department of Pathology Anatomy, Faculty of Medicine, University of Muslim Indonesia, Jl. Urip Sumoharjo KM.5, Makassar, Sulawesi 90231, Indonesia; 3Department of Neurobiology, Key Laboratory of Craniocerebral Disease, Ningxia Medical University, 1160 Shengli St, Xingqing District, Yinchuan 750004, Ningxia, China; 4Department of digestive and general surgery, Faculty of Medicine, Shimane University, 89-1 Enya, Izumo, Shimane 693-8504, Japan; ytajima@med.shimane-u.ac.jp; 5Department of Clinical Research, National Hospital Organization Kanmon Medical Center, 1-1 Chofu-Sotoura, Shimonoseki, Yamaguchi 752-8510, Japan

**Keywords:** Epstein–Barr virus, *Helicobacter pylori*, rapid urease test, EBV-associated gastric cancer

## Abstract

Persistent gastric mucosal damage caused by *Helicobacter pylori* infection is a major risk factor for gastric cancer (GC). The Epstein–Barr virus (EBV) is also associated with GC. Most patients with EBV-associated GC are infected with *H. pylori* in East Asia. However, very few reports have described where and when both *H. pylori* and EBV infect the gastric mucosa. To clarify this, old biopsy samples used for the rapid urease test (RUT) were applied to count EBV genomic DNA (gDNA) copies using DNA probe quantitative polymerase chain reaction. DNA extracted from the gastric biopsy samples of 58 patients with atrophic gastritis was used to analyze the correlation between the degree of atrophic gastritis and the copy number of EBV gDNA. EBV was detected in 44 cases (75.9%), with viral copy numbers ranging from 12.6 to 4754.6. A significant correlation was found between patients with more than 900 copies of EBV gDNA and those with a more severe grade of atrophic gastritis (*p* = 0.041). This study shows that EBV can be detected in RUT samples in a manner that reduces patient burden.

## 1. Introduction

*Helicobacter pylori* infection causes chronic active gastritis, atrophic gastritis, and gastric cancer (GC) [1]. The Epstein–Barr virus (EBV) is associated with 10% of all GC cases. Patients with EBV-associated GC (EBVaGC) primarily suffer from *H. pylori* gastritis [2,3]. Hatakeyama et al. reported that EBV infection induces methylation of the *Src homology region 2 domain-containing phosphatase-1* (*SHP1*) phosphatase gene promoter. The resulting SHP1 downregulation upregulates oncogenic *H. pylori* cytotoxin-associated gene A (CagA) phosphorylation [4]. Conversely, *H. pylori* CagA protein stimulates EBV-mediated epigenetic modifications and cell proliferation in a coinfection model [5]. Also, several cases show that the preceding infection of *H. pylori* promotes the development of EBVaGC [6]. We found that EBVaGC is usually present in the gastric body close to the border between the atrophic and normal gastric mucosa [7]. These findings indicate an association between *H. pylori* and EBV in the occurrence of GC [8]. 

Because EBV episomes in GC cells are a clonal propagation of the same ancestral virus, EBV infection is thought to be carcinogenic to gastric epithelial cells [9]. EBV infection suppresses apoptosis and increases cell motility [3]. However, it is difficult to perform a gastric biopsy as a clinical test to confirm EBV infection in gastric epithelial cells. For this reason, we performed a preliminary study to detect EBV by reusing the sections to be discarded after the rapid urease test (RUT), which is usually performed in clinical practice [10]. Here, we have increased the scale of the study and assessed the specificity and sensitivity of EBV detection.

In order to detect viral genomic DNA (gDNA) from human samples, several targeting genes have been reported using quantitative polymerase chain reaction (qPCR). Among them, there is *Bam*H I fragment A leftward frame 5 (BALF5), which is highly conserved in EBV strains [11], the dyad symmetry region (DSR) in the *ori*P sequence [12], and *Bam*H I fragment W (*Bam*H I W), which has repeated sequences and multiple target sites [13]. Because the affinity of each primer to the target sequence and quality of DNA from tissue samples affect the efficiency of gene amplification, we evaluated each primer set to establish the most sensitive and accurate qPCR assay.

Our previous study utilizing more than 10 tissue samples to detect EBV infection was not effective because the assay used a number of valuable gastric biopsy samples [13]. Furthermore, we failed to show where and when EBV infected the mucosal epithelia. Therefore, a less invasive and more sensitive assay is required.

EBV gDNA was detected in two biopsy sections after RUT. This application is less invasive to patients because samples are easily obtained from routine clinical examinations. Moreover, since our proposal detects both EBV and *H. pylori* from the same sample, quantitative data of EBV and *H. pylori* can be compared with the degree of clinical atrophic gastritis. This new evaluation system will enable clinicians to diagnose EBVaGC before patients present clinical manifestations.

## 2. Materials and Methods 

### 2.1. Sample Materials

Human gastric biopsy samples were obtained from 58 patients with upper gastrointestinal symptoms and negative histories of antibiotic therapy against *H. pylori* infection from 2017 to 2020 (Table 1). The endoscopy and RUT examinations were performed at the National Hospital Organization Kanmon Medical Center. All human samples were obtained with informed consent and managed according to the ethical rules of the Kanmon Medical Center and Shimane University.

Murine stomach tissue was collected from 16-week-old male BALBc mice. The animal experiment was conducted according to the ethical rules of Shimane University.

Human gastric epithelial AGS cells enhanced with green fluorescence protein, (eGFP)-EBV-infected AGS (AGS-EBV) cells, eGFP-EBV-infected MKN28 (MKN28-EBV) cells, and Burkitt lymphoma-derived Raji cells [14] were cultured in RPMI-1640 medium supplemented with 10% fetal bovine serum (Sigma-Aldrich, St. Louis, MO, USA), 100 U/mL penicillin, and 100 μg/mL streptomycin (Nacalai tesque, Kyoto, Japan), and were maintained at 37 °C and 5% CO_2_. 

### 2.2. Atrophic Gastritis Diagnosis

Atrophic gastritis was identified by endoscopic observation and classified based on the extension of the atrophic border of gastritis according to the Kimura–Takemoto classification [15]. Using endoscopy, the diagnosis of atrophic gastritis was confirmed with the visibility of vascular patterns, followed by the loss of gastric mucosal glands. According to the Kimura–Takemoto classification, closed-type atrophic gastritis (C1-3) was identified when the atrophic border remained on the lesser curvature of the stomach. If it extended along the anterior and posterior walls of the stomach, it was grouped into open-type GC (O1-3). Each of these cases was regrouped into mild (C1-2), moderate (C3-O1), and severe (O2-3) atrophic gastritis (Table 1).

### 2.3. H. Pylori Detection by RUT

*H. pylori* infection in human biopsy samples was evaluated using RUT. Biopsy samples were taken from two different suspected lesions of the gastric mucosal surface and RUT was performed using a PyloriTek test kit (Serim Research Corp, Elkhart, IN, USA). After testing, the specimens were stored at −80 °C until use.

### 2.4. gDNA Preparation

The gDNAs from human biopsy samples, murine tissue, and cell line samples were extracted using a GenElute^TM^ Mammalian Genomic DNA Miniprep Kit (Sigma-Aldrich), according to the manufacturer’s instructions. Briefly, tissues were thawed and weighed to up to 25 mg then cut into small pieces. Cell pellets were prepared that contained up to 5 × 10^5^ cells. Tissue samples were digested with lysis solution T, followed by Proteinase K buffer. The cell sample was resuspended in Resuspension Solution with Proteinase K and mixed with lysis solution C. After complete digestion, lysed solutions were mixed with ethanol and transferred into a binding column. When all flow-through liquids were dropped, the binding column was washed twice and the gDNA was eluted. The quantity and quality of the extracted gDNA was evaluated using a NanoDrop One spectrophotometer (Thermo Fisher Scientific, Waltham, MA, USA).

### 2.5. qPCR Targeting EBV gDNA

The DNA probe qPCR assay was used to determine the EBV copy number. Primers (*BamH* I W F: 5′- CCCAACACTCCACCACACC -3′, R: 5′- TCTTAGGAGGCTGTCCGAGG -3′ [16], BALF5 F: 5′- CTGACAAGGAGTACCTGCGT -3′, R: 5′- GAATGACGGCGCATTTCTCG -3′ [11], DSR F: 5′- ATGTAAATAAAACCGTGACAGCTCA -3′, R: 5′- TTACCCAACGGGAAGCATATG -3′ [12], glyceraldehyde 3-phosphate dehydrogenase (GAPDH) gene F: 5′- TGTGCTCCCACTCCTGATTTC -3′, R: 5′- CCTAGTCCCAGGGCTTTGATT -3′ [17], and eGFP gene F: 5′- GACAACCACTACCTGAGCAC -3′, R: 5′- C AGGACCATGTGATCGCG -3′ [18]), and double-quenched fluorescent DNA probes (*Bam*H I W: 5′- FAM/CACACACTA/ZEN/CACACACCCACCCGTCTC /IBFQ -3′, BALF5: 5′- FAM/CGGTCACAA /ZEN/TCTCCACGCTG/IBFQ -3′, DSR: 5′- FAM/CGAATTAGG/ZEN/CTTAGTAAAATGGTCC/ IBFQ -3′, GAPDH: 5′- FAM/CGGTCACAA/ZEN/TCTC CACGC/IBFQ -3′, eGFP: 5′- FAM/CCTGAGCAA/ZEN/GACCCCAACGAGAA/IBFQ -3′) were obtained from Integrated DNA Technologies (IDT, Coralville, IA, USA). The qPCR was performed in a 10 µL reaction volume containing 1 µL of gDNA template, 1 µL of 10 µM primers, 5 µL of 2X SSO Advanced Universal Probes Supermix (Bio-Rad, Hercules, CA, USA), and 1 µL of 5 µM DNA probe. The thermocycling conditions were as follows: 98 °C for 2 min, 95 °C for 10 s, and 60 °C for 30 s for 40 cycles. The reaction was detected using the CFX Connect Real-Time PCR Detection System (Bio-Rad). GAPDH served as an internal standard and for viral load normalization in each experiment. A standard calibration curve of the EBV genome was constructed from Raji and AGS-EBV gDNA cell lines, which had 50 and 180 copies of EBV per cell, equal to 27.3 × 10^6^ copies of EBV DNA per microgram.

### 2.6. Statistical Analysis

Categorical data distributions between groups were compared with the Chi-square test, Fisher exact test, and independent t-test using IBM SPSS Statistics 21.0 (SPSS, Chicago, IL, USA). Samples with more than 100 EBV gDNA copy numbers per μg of DNA were selected and analyzed via box-and-whisker plots. For all tests, *p*-values < 0.05 were considered statistically significant.

## 3. Results

### 3.1. Detection of EBV gDNA Using qPCR

gDNA from AGS, AGS-EBV, MKN-EBV, and Raji cells was used to assess the specificity and sensitivity of the DNA probe qPCR for EBV gDNA detection. The *Bam*H I W, DSR, and BALF5 probes did not react with the gDNA from EBV-negative AGS cells (Figure 1a). The differences in sensitivity for DNA primers and probes are shown in Figure 1b. Through AGS and MKN28 cells, *Bam*H I W, DSR, and BALF5 genes were detected in EBV-positive but not EBV-negative cells. Relative expression of the BALF5 and *Bam*H I W genes increased two-fold in AGS-EBV cells compared to in MKN28-EBV cells. These differences were compatible with the differences in the expression of eGFP control. However, the relative expression of DSR showed a four-fold increase in AGS-EBV cells compared to MKN-EBV cells (T-test *p* = 0.021). The intensity of *Bam*H I W gene expression was significantly enriched.

The sensitivity of qPCR was confirmed under various conditions by diluting AGS-EBV cell-derived gDNA with murine stomach gDNA. The gDNA consisted of foods and other bacteria, in addition to murine tissue. We performed several qPCR assays to estimate the condition to be tested with human samples. Accordingly, we used this murine gDNA to assume the gDNA obtained from biopsy samples. As a result, *Bam*H I W showed higher sensitivity (y = 0.9157x + 0.1324, R^2^ = 0.9518) than BALF5 (y = 0.1905x + 0.0408, R^2^ = 0.9404) (Figure 1c). As the *Bam*H I W primers and probe showed high specificity and sensitivity, we used qPCR for *Bam*H I W for subsequent research.

To examine the specificity of the primer, gDNA from AGS-EBV cells was diluted and a calibration curve for luminescence was prepared (Appendix A). The specificity of the primer was high enough to show R^2^ at 0.9995. We have also examined primer efficiency by creating a plasmid that contains *Bam*HI W EBV DNA fragment. We used the plasmid DNA as a template and created another calibration curve for luminescence (Appendix A). The specificity of the primer showed R^2^ at 0.9999.

In addition, we assessed the EBV copy number in gDNA from AGS-EBV and MKN28-EBV cells by comparing them with Raji gDNA, which contains 50 episomal EBV copies per cell. AGS-EBV and MKN28-EBV cells contained 180 and 100 EBV copies per cell, respectively (Figure 1d). 

### 3.2. EBV gDNA Load in Atrophic Gastritis

Of the 58 patients, 27 (46.6%), 25 (43.1%), and 6 (10.3%) were diagnosed with mild, moderate, and severe atrophic gastritis, respectively (Table 1). Patients with mild gastritis were younger than patients with moderate or severe gastritis (total 56.78 ± 15.87; 68.80 ± 10.37; and 66.33 ±13.35; *p* = 0.003 vs. 0.65, t-test). Gender differences did not affect either EBV positivity (*p* = 0.6) or atrophic gastritis grade (*p* = 0.27). *H. pylori* infection was detected in 44 cases (75.9%) from gastric mucosa samples using RUT and in 14 cases (24.1%) from RUT-negative samples (Table 1).

EBV gDNA was detected by qPCR assay in 44 samples (75.9%) but not in 14 samples (24.1%). The EBV copy number per 1 µg of gDNA was lower than 100 in 10 samples (17.2%), higher than 100 in 34 samples (58.6%), and higher than 1000 in 9 samples (15.5%) (Figure 2a). Nine cases with a copy number higher than 1000 are indicated with an arrow. The EBV copy number was plotted in each of the three groups divided by the severity of atrophic gastritis (Figure 2b). The median copy numbers of EBV per µg of sample gDNA were 515, 933, and 349 for mild, moderate, and severe atrophic gastritis, respectively. Samples with a copy number higher than 900 EBV per microgram of gDNA were frequently observed in moderate atrophic gastritis (*p* = 0.041).

## 4. Discussion

*H. pylori* infection can cause gastritis and GC, and EBV infection can also cause GC. We examined 58 patients with gastritis and detected *H. pylori* infection in 44 samples by RUT (Table 1). RUT is commonly used to diagnose *H. pylori* infection due to its high sensitivity (85–100%) and specificity (100%) in detection [19,20]. However, at least 10^5^
*H. pylori* are required to obtain positive results by RUT [21]. Therefore, we performed nested PCR on five RUT-negative cases. The results confirmed that all negative cases were positive for *H. pylori* (data not shown). Our results were similar to those of a previous study wherein 30% of RUT-negative patients became positive after a specific PCR assay [19].

We compared the specificity and sensitivity of *BALF5*, *DSR*, and *BamH I W* to determine the most appropriate target for EBV detection. Any set of primers and probes for the three genes did not react with gDNA from EBV-negative human gastric cells. The specificity of primers and probes was also confirmed using murine stomach gDNA. There was no nonspecific reaction in murine stomach gDNA, but murine stomach gDNA mixed with gDNA from AGS-EBV cells showed positive results. The *BamH I W* gene showed the best specificity and sensitivity among the three genes (Figure 1), probably because *BamH I W* genes consist of repeated sequences.

The qPCR assay targeting the *BamH I W* gene was performed to determine EBV copy numbers in 58 post-RUT samples. The double quencher DNA probe used will provide more specificity and sensitivity than a single probe because the nonspecific signal from one of the quenchers will be suppressed by the other unbound quencher. The quality of the tissue sample was seemingly substandard as it may have contained food or bacteria. Moreover, the EBV genome may be destroyed because RUT samples are exposed to room temperature without any preserving efforts during RUT examination. However, by utilizing a DNA template from two RUT samples per case, we detected the EBV genome in 44 of the 58 samples (76%) (Figure 2a). We detected *H. pylori* DNA in five RUT-negative cases, which may be reflected in the small number of *H. pylori* infections.

We showed that most patients who were EBV-infected had moderate atrophic gastritis. Statistical analysis revealed that the median EBV DNA copy numbers per microgram of gDNA from mild, moderate, and severe atrophic gastritis samples were 515, 933, and 349, respectively. Moreover, samples with a copy number higher than 900 EBV were frequently observed in moderate atrophic gastritis (*p* = 0.041) (Figure 2b). Hirano et al. also used the *Bam*H I W fragment and detected both EBV and *H. pylori* infections in patients with moderate grade atrophic gastritis [13]. They needed at least 10 fresh gastric tissue specimens to detect EBV, probably because they used a less sensitive single quencher probe.

Here, we established a method for detecting EBV from RUT samples, which will assist in the study of large numbers of *H. pylori* gastritis cases. Reuse of RUT specimens reduces the invasiveness of repeated gastroscopy in patients. Using only two biopsy sections after RUT, we frequently detected EBV in mucosa with moderate atrophic gastritis. The result is similar to or better than the previous study that detected EBV in the atrophic mucosa [22]. EBV may be transmitted to the mucosa suffering from *H. pylori* gastritis in a similar fashion to the experimental cell-to-cell transfer of EBV to gastric epithelial cells. Therefore, monitoring the carcinogenic progress in clinical EBVaGC cases will help elucidate the pathological traits for the remaining 90% of other types of GC. We will clarify the dynamics of EBV infection in human gastric mucosa using this system.

There are some limitations to our study. First, patients who received RUT were more likely to become *H. pylori*-positive, which does not reflect the infection status in the normal population. Therefore, we aim to investigate more *H. pylori*-negative cases. Secondly, whether the detected EBV signal was derived from epithelial cells or infiltrated B lymphocytes is still unclear. The combination with histological detection of EBV-specific signals will provide further information regarding the pathogenesis. However, collecting additional fresh biopsy samples is costly and can cause physical distress to the patients. Third, though EBV copy number was detected in moderate patients and severe patients, there are also many mild patients with higher EBV copy number compared to moderate and severe patients. This underlines EBV copy number may not be the sole determinant driving EBVaGC. Various host factors, variable viral strains, and difference in lifestyle may affect tumorigenesis. Thus, observing the process from gastritis to gastric cancer formation will be useful in clarifying the pathophysiology of EBVaGC. Finally, because the number of patients analyzed in the study was small, we will keep collecting samples to get more accurate statistical data.

## 5. Conclusions

The findings of this study showed that this DNA probe qPCR assay is highly sensitive for detecting EBV infection in RUT samples. Applying this method to a large number of samples will assist in proving its efficacy and efficiency. By using two biopsy specimens, we discovered the presence of EBV together with *H. pylori*. Therefore, employing this method to detect EBV infection in gastric mucosa during early-stage gastritis will be helpful for preventing EBVaGC.

## Figures and Tables

**Figure 1 microorganisms-08-00923-f001:**
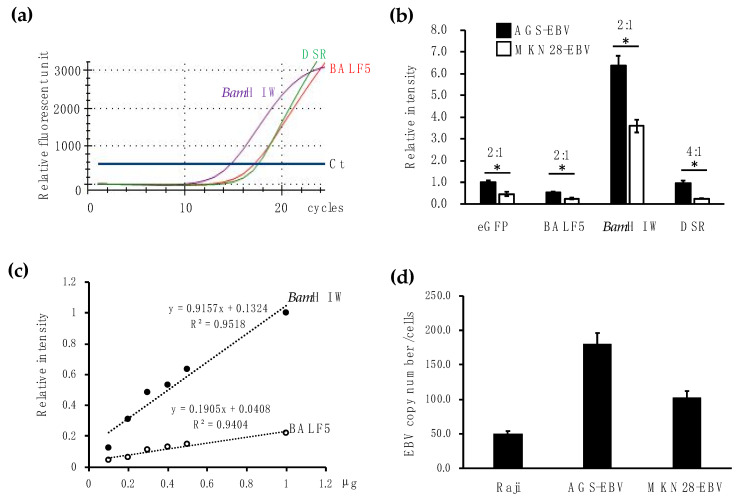
Detection of EBV DNA using qPCR for different targets. (**a**) The qPCR DNA probe reacted with gDNA from AGS-EBV cells but not with that from AGS cells. Signals using the *BamHI* W (purple line), BALF5 (red line), and DSR (green) probes are indicated. Blue line: Ct value. (**b**) Difference in signal intensity (sensitivity) for detecting EBV gDNA between two DNA probes. eGFP was used as a control. Filled black and open squares indicate results using gDNA obtained from AGS-EBV and MKN28-EBV cells, respectively. * *p* < 0.05. (**c**) Specificity and sensitivity of *Bam*HI W (closed circle) and BALF5 (open circle) probes to detect individual EBV genes. gDNA from AGS-EBV cells diluted with that from murine stomach was used. (**d**) Detection of EBV copy number using the *BamHI* W probe from the gDNA of AGS-EBV and MKN28-EBV cells. The gDNA from Raji cells containing 50 episomal EBV copies per cell was used as a standard.

**Figure 2 microorganisms-08-00923-f002:**
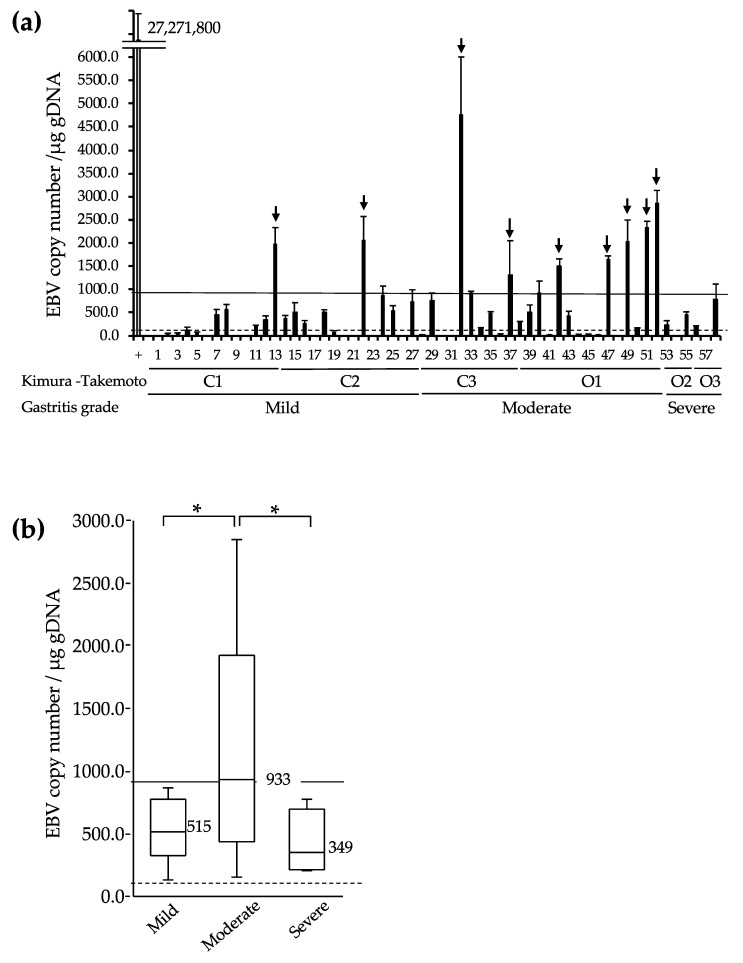
Detection of EBV gDNA from RUT biopsy samples. (**a**) EBV gDNA was detected using a *BamHI* W probe from RUT samples. Arrows indicate cases with more than 1000 EBV copies per µg of gDNA. AGS-EBV cell marked + (grey bar) and AGS cell marked–have 27,271,800 and 0 EBV copies per µg of gDNA, respectively. Code numbers from 1 to 58 are RUT samples (black bars). (**b**) Correlation between EBV copy number and grade of atrophic gastritis. The vertical bars outside the boxes indicate distributions. Numbers in the center of the bars indicate median EBV copy numbers in each atrophic grade. The solid and dotted lines indicate 900 and 100 EBV copies per microgram of gDNA, respectively. Samples with less than 100 EBV copies per microgram of gDNA were excluded from the box-and-whisker plot analysis due to poor reproducibility of calculations. * *p* < 0.05.

**Table 1 microorganisms-08-00923-t001:** Characteristics of endoscopic biopsy samples.

Characteristic	Biopsy Samples (*N* = 58)
Gender	
Female (%)	25 (43.1)
Male (%)	33 (56.9)
Median age (years) (range)	63 (26–93)
Atrophic gastritis grade (Kimura–Takemoto classification)	
C1 (mild) (%)	13 (22.4)
C2 (mild) (%)	14 (24.1)
C3 (moderate) (%)	10 (17.2)
O1 (moderate) (%)	15 (25.9)
O2 (severe) (%)	3 (5.2)
O3 (severe) (%)	3 (5.2)
*H. pylori* infection (RUT)	
*H. pylori* positive (%)	44 (75.9)
*H. pylori* negative (%)	14 (24.1)

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
