# Peer review of "Application of Biopsy Samples Used for Helicobacter pylori Urease Test to Predict Epstein–Barr Virus-Associated Cancer"

_microorganisms, 2020, doi:10.3390/microorganisms8060923_

Round 1

Reviewer 1 Report

It is well established that Epstein-Barr virus (EBV) and Helicobacter pylori (H. pylori) are two pathogens associated with the development of various human cancers. In addition, the coexistence of both microorganisms in gastric cancer specimens has been increasingly reported, implicating a possible cross-talk between the two pathogens during the oncogenic process.

In the manuscript "Application of biopsy samples used for Helicobacter pylori urease test to predict Epstein-Barr virus-associated cancer" by Andy Visi Kartika et al, the authors provide a neat prospect for every day clinical practice: the reuse of the same sections to detect H. pylori first with RUT, and then - for EBV detection with qPCR. This work is a logical follow up to the previous publication in Endosc. Int. Open, 2019, 7, E341-2. The authors now increase the scale of the experimental component by using more samples and assessing the specificity and sensitivity of EBV detection.

Generally, the manuscript is straightforward and easy to follow and understand. The last paragraph of the Discussion is quite refreshing on the point of critical understanding of certain limitations of the study. 

In summary, the implementation of the described approach to clinical practice will reduce the number of invasive procedures in patients with early stages of oncogenesis, which is tremendously vital for cancer prevention and early treatment.

Specific comment

It is quite evident that for some reason the authors do not discuss the latest data published on the correlation between Helicobacter pylori and EBV in gastric carcinomas (apart from their own preliminary study, the most recent publications on the reference list are from the spring of 2018). Information from the latest studies would make this work even more relevant.

Reviewer 2 Report

The patient number is small to support the results.  There are only 3 patients with severer symptoms included in the study, this has low n to support the conclusions.

Though EBV copy number was detected in moderate patients and severe patients (n=3). There are also many patients with higher EBV copy number in mild patients compared to moderate and severe patients. This underlines EBV copy number may not be the sole determinant driving the disease.

Reviewer 3 Report

Andy Visi Kartika and colleagues utilized samples from patients tested for H.Pylori associated cancer to test for EBV using PCR.

  1. The authors utilized probe-based qPCR assays. Were the primers tested for primer efficiency using the slope of the standard curve?
  2. The tm curve of the primers and primer dimers should also be considered in such testing and should be included in supplemental
  3. The EBV copy number as it is stated in the paper seems to be determined by the assumption that Raji cells contain 50 copies. This reviewer would like to see a standard curve using a known well-charachterized plasmid that encodes EBV gene that is being tested to plot a standard curve and calculate the molar amount of DNA that correlates with the copy number.
  4. Finally, a lot of this is the only qPCR. At least for a few samples, this reviewer suggests ISH/FISH for EBERS or antibody-based tests to show the presence of EBV.

Round 2

Reviewer 2 Report

NA

Author Response

Thank you very much for providing us valuable comments.

Q1. The patient number is small to support the results. There are only 3 patients with severer symptoms included in the study, this has low n to support the conclusions.

A1. We are attempting to increase the number of sample sizes. After the initial submission, we collected additional 34 samples, which were 10 mild, 10 moderate, and 3 severe gastritis cases including with 11 H. pylori-negative cases. Three newly obtained patient samples with severe gastritis were added to the three old samples and compared with the patient sample with moderate gastritis, which showed that the EBV copy number was significantly higher in moderate gastritis (p<0.05). In order to minimize uncertainty, additional discussions to mention the limitation in terms of small sample sizes of the current study were made in lines from 247 to 252.

Reviewer 3 Report

Although, I am satisfied with the progress made. I think, it is best that this manuscript comes with a proper number of sample sizes as pointed out by other reviewers as well.

Author Response

 Thank you very much for reviewing. Our collective responses are as follows:

Q1. I am satisfied with the progress made.

A1. We are very thankful to your valuable comments. We got chances to confirm our methodology and techniques are confirmative. We are now more convincing the accuracy of the results.

Q2. It is best with this manuscript comes with a proper number of sample sizes as pointed out by other reviewers as well.

A2. We are attempting to increase the number of sample sizes. After the initial submission, we collected additional 34 samples, which were 10 mild, 10 moderate, and 3 severe gastritis cases including with 11 H. pylori-negative cases. Three newly obtained patient samples with severe gastritis were added to the three old samples and compared with the patient sample with moderate gastritis, which showed that the EBV copy number was significantly higher in moderate gastritis (p<0.05). In order to minimize uncertainty, additional discussions to mention the limitation in terms of small sample sizes of the current study were made in lines from 247 to 252.

We appreciated all of the helpful suggestions from the referees. We hope the revised manuscript will meet your standards and can be accepted for publication in the special issue of Microorganisms

Round 3

Reviewer 2 Report

NA

Author Response

Thank you very much for reviewing. We agree with the opinion to revise our manuscript including additional sample data. We have added data from another 34 samples, which were 10 mild, 10 moderate, and 3 severe gastritis cases including 11 H. pylori-negative cases. We modified our manuscript according to the increase in sample size. The followings are details by correcting previous description with a horizontal bar and indicating modifications as bold characters:

  • Line 28 (Abstract):

Sentences “DNA extracted from the gastric biopsy samples of 33 patients with atrophic gastritis ……” have been changed to “DNA extracted from the gastric biopsy samples of 58 patients with atrophic gastritis ……”.

  • Lines from 29 to 30 (Abstract):

Sentences “EBV was detected in 27 cases (82%), with viral copy numbers ranging from 12 to over 1,000.” havebeen changed to “EBV was detected in 47 cases (81%), with viral copy numbers ranging from 12 to over 4,756.”.

  • Lines from 30 to 32 (Abstract):

Sentences “A significant correlation was found between patients with more than 750 copies of EBV gDNA and those with a more severe grade of atrophic gastritis (p = 0.046).” have been changed to “A significant correlation was found between patients with more than 900 copies of EBV gDNA and those with a more severe grade of atrophic gastritis (p = 0.041).”.

  • Lines from 72 to 73 (Materials and Methods):

The sentence “Human gastric biopsy samples were obtained from 33 patients with upper gastrointestinal symptoms and negative histories of antibiotic therapy against H. pylori infection from 2017 to 2019.” have been changed to “Human gastric biopsy samples were obtained from 58 patients with upper gastrointestinal symptoms and negative histories of antibiotic therapy against H. pylori infection from 2017 to 2020.”.

  • Line 77 (Materials and Methods):

Numbers in Table 1 have been changed according to the increase in sample numbers.

  • Line 170 (Results):

The sentences “Of the 33 patients, 16 (48.5%), 14 (42.4%), and three (9.1%) were diagnosed with ……..” have been changed to “Of the 58 patients, 27 (46.6%), 25 (43.1%), and 6 (10.3%) were diagnosed with ……..”.

  • Lines from 171 to 173 (Results):

The sentences “There was no age difference among patients with mild, moderate, or severe gastritis (total 55.63 +15.23; 65.36 + 9.30; and 58.00 + 15.00; p = 0.057 vs. 0.64, t-test).” have been changed to “Patients with mild gastritis were younger than patients with moderate or severe gastritis (total 56.78 + 15.87; 68.80 + 10.37; and 66.33 + 13.35; p = 0.003 vs. 0.65, t-test).”.

  • Lines from 173 to 175 (Results):

The sentences “No gender difference was found among them (p = 0.87).” have been changed to “Genderdifferences did not affect either EBV positivity (p = 0.6) or atrophic gastritis grade (p = 0.27).”.

  • Lines from 174 to 175 (Results):

The sentences “H. pylori infection was detected in 28 cases (84.8%) from gastric mucosa samples using RUT and in five cases (12.2%) from RUT-negative samples.” have been changed to “H. pylori infection was detected in 44cases (75.9%) from gastric mucosa samples using RUT and in 14 cases (24.1%) from RUT-negative samples.”.

  • Line 176 (Results):

The sentences “EBV gDNA was detected by qPCR assay in 27 samples (81.8%) but not in six samples (18.2%).” have been changed to “EBV gDNA was detected by qPCR assay in 44 samples (75.9%) but not in 14 samples (24.1%).”

  • Lines from 177 to 178 (Results):

The sentences “The EBV copy number per 1 µg of gDNA was lower than 100 in 10 samples (30.3%), higher than 100 in 17 samples (51.5%), and higher than 1,000 in two samples (6.1%) (Figure 2a).” have been changed to “The EBV copy number per 1 µg of gDNA was lower than 100 in 10 samples (17.2%), higher than 100 in 34 samples (58.6%), and higher than 1,000 in nine samples (15.5%) (Figure 2a).”

  • Lines from 178 to 179 (Results):

The sentences “Two cases with a copy number higher than 1,000 are…...” have been changed to “Nine cases with a copy number higher than 1,000 are ……..”

  • Lines from 180 to 181 (Results):

The sentences “The median copy numbers of EBV per µg of sample gDNA were 480, 762, and 349 for mild…...” have been changed to “The median copy numbers of EBV per µg of sample gDNA were 515, 933, and 349 for mild…..”

  • Lines from 182 to 183 (Results):

The sentences “Samples with a copy number higher than 750 EBV per microgram of gDNA were frequently observed in moderate atrophic gastritis (p = 0.046).” have been changed to “Samples with a copy number higher than 900 EBV per microgram of gDNA were frequently observed in moderate atrophic gastritis (p = 0.041).”

  • Line 188 (Legend of Figure 2):

The sentence “Code numbers from 1 to 33 are RUT samples.” has been changed to “Code numbers from 1 to 58are RUT samples.”

  • Line 191 (Legend of Figure 2):

The sentence “The solid and dotted lines indicate 750 and 100 EBV copies” has been changed to “The solid and dotted lines indicate 900 and 100 EBV copies”

  • Lines from 195 to 196 (Discussion):

The sentence “We examined 33 patients with gastritis and detected H. pylori infection in 28 samples by RUT.” has been changed to “We examined 58 patients with gastritis and detected H. pylori infection in 44 samples by RUT.”

  • Lines from 216 to 217 (Discussion):

The sentence “However, ……, we detected the EBV genome in 27 of the 33 samples (81.8%) (Figure 2a).” has been changed to “However, ……, we detected the EBV genome in 44 of the 58 samples (76%) (Figure 2a).”

  • Lines from 220 to 221 (Discussion):

The sentence “Statistical analysis revealed that ….. and severe atrophic gastritis samples were 480, 762, and 349, respectively.” have been changed to “Statistical analysis revealed that ….. and severe atrophic gastritis samples were 515, 933, and 349, respectively.”.

  • Lines from 221 to 223 (Discussion):

The sentence “Moreover, samples with a copy number higher than 750 EBV were frequently observed in moderate atrophic gastritis (p = 0.046) (Figure 2b).” have been changed to “Moreover, samples with a copy number higher than 900 EBV were frequently observed in moderate atrophic gastritis (p = 0.041) (Figure 2b).”.

  • Lines from 249 to 252 (Discussion):

Because we have added new data and modified revised manuscript, we deleted sentences “Three newly obtained patient samples with severe gastritis were added and compared with the patient sample with moderate gastritis, which showed that the EBV copy number was significantly higher in moderate gastritis (data not shown). Therefore, we plan to show more accurate results in the next paper.” in the revised manuscript.

We appreciated all of the helpful suggestions from the referees. We hope the revised manuscript will meet your standards and can be accepted for publication in the special issue of Microorganisms

Reviewer 3 Report

Since authors have already collected more samples, this manuscript should be revised with all collected sample sizes and not just with 3 samples. Therefore, they should resubmit the manuscript with an increased sample size.

Author Response

(The authors gave the same response as above.)

Round 4

Reviewer 3 Report

I am satisfied with the changes.